# Application of Immobilized Enzymes in Juice Clarification

**DOI:** 10.3390/foods12234258

**Published:** 2023-11-24

**Authors:** Feng Wang, Hui Xu, Miaomiao Wang, Xiaolei Yu, Yi Cui, Ling Xu, Anzhou Ma, Zhongyang Ding, Shuhao Huo, Bin Zou, Jingya Qian

**Affiliations:** 1School of Food and Biological Engineering, Jiangsu University, Zhenjiang 212013, China; 2222118040@stmail.ujs.edu.cn (H.X.); 2221918079@stmail.ujs.edu.cn (M.W.); 2222018099@stmail.ujs.edu.cn (X.Y.); cuiyi@ujs.edu.cn (Y.C.); lxu@ujs.edu.cn (L.X.); huo@ujs.edu.cn (S.H.); binzou2009@ujs.edu.cn (B.Z.); sjqjy@126.com (J.Q.); 2Institute of Agricultural Products Processing Engineering, Jiangsu University, Zhenjiang 212013, China; 3Research Center for Eco-Environmental Sciences, Chinese Academy of Sciences, Beijing 100085, China; azma@rcees.ac.cn; 4Key Laboratory of Carbohydrate Chemistry and Biotechnology, Ministry of Education, School of Biotechnology, Jiangnan University, Wuxi 214122, China; bioding@163.com

**Keywords:** immobilized enzyme, immobilization method, immobilized support, juice clarification, clarification mechanism

## Abstract

Immobilized enzymes are currently being rapidly developed and are widely used in juice clarification. Immobilized enzymes have many advantages, and they show great advantages in juice clarification. The commonly used methods for immobilizing enzymes include adsorption, entrapment, covalent bonding, and cross-linking. Different immobilization methods are adopted for different enzymes to accommodate their different characteristics. This article systematically reviews the methods of enzyme immobilization and the use of immobilized supports in juice clarification. In addition, the mechanisms and effects of clarification with immobilized pectinase, immobilized laccase, and immobilized xylanase in fruit juice are elaborated upon. Furthermore, suggestions and prospects are provided for future studies in this area.

## 1. Introduction

Fruit juices are among the food groups that are very common in our daily diet due to their pleasant taste and positive effects on the digestive system [1,2]. More and more consumers are aware of the relationship between the intake of biologically active compounds obtained from fruit juice and the maintenance of health. These ingredients can effectively prevent cardiovascular disease, various cancers, diabetes, etc.; thus, they are of great benefit to human health [1,3,4,5]. This has caused the global market for fruit juice to grow rapidly in the last few years, and the consumption of fruit juice increases rapidly day by day, showing broad prospects for future development [6]. In juice production and processing, the main production process includes washing, selection, pressing, separation and centrifugation, juice clarification, filtration, and so on; finally, the clarified juice is obtained (Figure 1) [7,8].

The clarification of fruit juice is a key step in the process of fruit juice production, and it is related to the quality, sensory qualities, and storage stability of the juice. At present, mainly physical methods, chemical methods, and enzymatic methods are used to clarify juice at home and abroad. The most commonly used physical methods include the method of adding clarifiers, for example, honey, albumin, etc. This mainly involves the use of some proteins in these clarifiers in combination with phenolic substances in the juice itself to form turbidity precipitates. After filtration, these precipitates can result in a clear and non-browning juice. In addition, adsorbents, such as PVPP (polyvinyl pyrrolidone), bentonite, and other substances with a large surface area and strong adsorption capacity, are also used for adsorption clarification [9,10], and most tannins and phenolic substances in juice are removed via adsorption. In addition, this also includes the freezing clarification method and the heating coagulation clarification method, which mainly use a change in temperature to make the gelatinous substances in the juice accumulate in order to filter and remove them. Commonly used chemical methods include the sulfite method, electrocoagulation method, ozonation, ion exchange method, etc. Chemical methods cause some environmental problems, resulting in changes in the nutritional qualities of the juice; in addition, the post-treatment requires the removal of impurities, which increases the operating cost. Physical methods have problems such as low clarification efficiency and difficult removal of the adsorbent. If some adsorbent remains in the juice, its quality will be affected. Moreover, the reactive phenolic compounds in the juice cannot be completely removed, resulting in unstable turbidity. Membrane technology is also used for juice clarification. Suspended slurry particles in the juice cause membrane pollution and damage, increasing the economic cost to a certain extent [11]. At present, the enzymatic method is widely used in juice clarification, and the related enzymes mainly include pectinase, laccase, and xylanase. These enzymes are considered sustainable and environmentally friendly industrial catalysts [12], and they are mainly derived from *Penicillium oxalate* F67 [13], *Bacillus discoloriformis* [14], and *Aspergillus flavus* [15]. However, most juice is acidic, and the conformation and structure of these enzymes will be affected by the strongly acidic pH, which will reduce their function and greatly limit their applications in juice. Therefore, acidic enzymes are generally used in juice clarification.

The industrial application of natural free enzymes is hindered due to their unstable operability and poor reusability [16,17,18,19,20]. In addition, the incorporation of free enzymes into fruit juice causes some other problems. Due to the good solubility of enzymes in liquid systems, free enzymes can remain in juice, and the catalyzed reaction may continue or be out of control, resulting in unstable juice quality. Since enzymes include food allergens [21], there are consumers who may be allergic to the enzymes remaining in juice. Immobilization is the process of confining enzymes in a defined space to convert the catalyst from homogenous (free enzyme) into heterogenous (immobilized enzyme) [22,23]. The technology of enzyme immobilization has developed rapidly in recent years. This technology involves the use of physical or chemical methods to immobilize free-form enzymes and maintain their biological activity [24] so that the enzymes can be easily recycled and reused. Proper immobilization can also improve the stability, activity, purity, selectivity, and specificity of an enzyme [25,26]. Immobilization can also protect an enzyme from adverse environmental conditions (such as high temperatures, extreme pH values, etc.) [27], prevent adverse conformation changes, and improve the enzyme’s function so that it can show high catalytic activity in a wide range of pHs and temperatures [28], in addition to improving its catalytic efficiency [29]. In addition, immobilized enzymes with excellent reusability and controllability have been developed in many studies over the last several decades; this has enabled enzymes to be better separated from reaction mixtures, thus improving their recycling, which is very critical and important in order to achieve great cost reductions in the juice clarification industry [29]. However, most published reviews and research articles have focused on the advantages of enzyme immobilization without highlighting its problems. In fact, there are still some problems, such as difficulties in scaling up immobilization for industrial applications, the high cost of biocatalysts, and the negative impact on biocatalyst performance [23]. A review by Bolivar et al. highlighted the shortcomings of immobilization schemes that use incorrect designs and described solutions to overcome these problems [23]. As catalytic enzyme properties are closely related to the enzyme conformation, it may be assumed that the activity, specificity, selectivity, and even inhibition of the immobilized enzyme could be altered [23]. Enzyme release may occur in reversible immobilization under certain circumstances, resulting in product contamination and a decrease in reactor productivity [23]. Strengthened interaction between enzyme and support could be achieved by different strategies, including crosslinking of the adsorbed enzyme or subunits in multimeric enzymes, hetero functional supports and polymeric beds [23,30]. The interaction between the enzyme and support may lead to the distortion of some enzymes [23], but it may produce an enzyme hyper-activation in certain cases [23,31]. Stronger multipoint covalent immobilization can be achieved through coupling site specific mutations of enzyme or chemical modifications of supports [30]. When enzymes were immobilized into porous solid supports, substrate access can be hindered due to the diffusion limitations of too large substrates or incorrect enzyme orientations in pores [23]. In these cases, the use of non-porous supports may solve this problem, where magnetic nanosupports can be a good alternative because they are easy to handle and recycle [23,32]. To relieve the mass transfer limitation resulting from product accumulation in the application of immobilized enzymes, enzyme cascades or co-immobilized enzymes exhibit a positive effect [23,33]. Therefore, once the immobilization is properly designed, different functional advantages can be obtained such as improved enzyme stability and reaction specificity, enhanced reusability, and better adaption to incompatible reaction conditions [23]. It was suggested that the enzyme immobilization process may have better results if the following three aspects are properly considered: appropriate scaffolds, appropriate active groups in the scaffolds, and appropriate immobilization regimens. It was also concluded that developments in the fields of material science, reactor engineering, protein science, organic chemistry, and biological process engineering may open new opportunities to solve these problems in the future. A review by Kotchakorn et al. [34] provided some new enzyme immobilization schemes based on materials science, where the progress of enzyme immobilization using organic–inorganic nanocrystals, metal–organic frameworks, and graphene-based nanomaterials was summarized and discussed. Recently, 3D printing technology has attracted researchers’ interest. Pose Boirazian et al. [35] reviewed the latest industrial and clinical applications of 3D printing for enzyme immobilization. Although there are still problems in enzyme immobilization, immobilized enzymes are widely used in the medical, food, detergent, textile, and pharmaceutical industries, and it is still necessary to explore novel immobilization strategies to meet the demands of industrial applications [27].

The advantages of immobilized enzymes have made them more and more popular in the clarification of fruit juice; they have been widely used in the clarification of apple, grape, citrus, pineapple, pomegranate, papaya, and other fruit juices and have shown superior clarification effects. In published review articles, the methods of immobilizing enzymes and their applications in food were reviewed. These reviews mainly summarized the immobilization strategies, the types and selection of immobilization supports, the application of immobilized enzymes in juice clarification, and their potential market prospects [36]. Another study focused on the market application of immobilized pectinase in juice clarification [37]. Some reviews also mentioned the use of immobilized enzymes to clarify fruit juices and discussed the best process parameters for clarifying fruit juices with immobilized enzymes and the changes in juice transmittance, acidity, and other parameters [38].

This review mainly summarizes the methods of immobilizing enzymes and their application in juice clarification, and it discusses the catalytic conditions and selectivity of immobilized enzymes in juice clarification, as well as the clarification mechanisms of commonly used immobilized enzymes and the clarification effects achieved in juice. 

## 2. Immobilized Enzymes in Juice Clarification

### 2.1. Immobilization Methods

Enzyme immobilization was defined as a sustainable approach [39,40] that results in the restriction or confinement of enzymes within a matrix or on the surface of a support through chemical or physical interactions without disrupting catalytic activity [41]. When applied to juice clarification, enzyme immobilization is considered to be one of the best techniques for improving enzyme activity, stability, and reusability [42]. At present, there are many methods for immobilizing enzymes. Because different enzymes have different characteristics, their immobilization supports and immobilization methods are not the same. In addition, some enzymes have more than one immobilization method, and some enzymes need a combination of a variety of methods for their immobilization. The methods of enzyme immobilization used in juice clarification mainly include adsorption, entrapment, covalent attachment, and cross-linking.

Each strategy has its advantages and disadvantages. At present, the simplest method is the use of physical adsorption on existing solids. Adsorption is a reversible immobilization method that usually does not change the inherent structure of an enzyme [43,44]. However, since the interaction between the enzyme and the support is weak with this method, the enzyme can be leached during the reaction [45]. In addition, the adsorption of some components of the juice could occur, and this may affect the quality of the juice. Entrapment is the retention of enzymes in a network structure, which prevents the aggregation and leaching of enzymes [46]. However, this method only allows small molecules to pass through, which causes limited mass transfer and low enzyme loads [47]. Covalent binding can bind an enzyme firmly to the support, thus avoiding enzyme shedding [43]. However, covalent bonds are irreversible [46], and strong binding forces during modification may destroy the active conformation of the enzyme, thereby reducing its catalytic activity [45]. Cross-linking is the use of bifunctional cross-linking reagents to link enzyme molecules [46], and cross-linked enzyme aggregates (CLEAs) or crystals (CLECs) are the most commonly used methods [44]. These methods can improve the performance of an enzyme. However, cross-linking agents can change the conformation of the enzyme, resulting in a loss of enzyme activity [46]. There is currently no universal immobilization strategy that can be used to address all limitations in various enzymatic processes. Before choosing which method to use, it is necessary to consider the properties of an enzyme, the process of the method, and the properties of the substrate.

#### 2.1.1. Adsorption

In the adsorption method, enzymes are immobilized on supports via van der Waals interactions, electrostatic adsorption, and/or hydrophobic interactions [46]. There are two kinds of adsorption methods: physical adsorption and ion adsorption [42]. Immobilization through physical adsorption method is a simple and rapid method [41]. At present, organic and inorganic porous materials, such as kaolinite, diatomite [47,48], silica gel pumice [49], magnetic corn starch microspheres [13], and cellulose [50], are the most common immobilized supports for physical adsorption. Sahin et al. [49] used Zr-treated pumice to immobilize pectinase, and the adsorption capacity reached 229 mg/g of support. The immobilized pectinase showed good thermal stability and reusability in juice clarification. Chauhan et al. [48] used diatomite to adsorb and immobilize pectinase, and an adsorption capacity of 20 mg pectinase/g diatomite was achieved. Compared with those of the free enzyme, the reaction temperature and thermal stability were improved, and excellent results were obtained in the clarification of pineapple juice.

Ionic adsorption is an immobilization method in which an enzyme is bound to a non-water-soluble support with ion-exchange groups through ionic binding [51]. The supports used for ionic adsorption include agarose, chitosan, polyethylene-imine, and epoxy-activated acrylate copolymer [49,50]. It was reported that immobilized pectinase on polyethyleneimine and epoxy-activated acrylate copolymer showed good tolerance and catalytic activity in a wide pH range of 3–7, and the immobilized pectinase retained 95% of its initial activity after being recycled more than 10 times in the process of juice clarification [49]. 

#### 2.1.2. Entrapment

Entrapment is a physical immobilization method, and it mainly involves two types of processes: gel entrapment and micro-entrapment. The principle is that an enzyme is embedded in an entrapping material with high density porosity and a high specific pore volume, and a permeable membrane that allows small molecular substrates to enter the material and react with the enzyme is formed on the surface. After the reaction, the metabolites can pass through the membrane, and the enzyme can be retained well in the network structure. At present, common entrapment materials include natural gels (e.g., chitosan beads), alginate, gelatin, and agarose gels, and synthetic gels (e.g., polyacrylamide and polyvinyl alcohol) [50]. A polyvinyl alcohol (PVA) sponge [14], agarose [52], polyacrylamide (PAM), and alginate [53,54] are commonly used in research on juice clarification. Esawy et al. [14] studied the immobilization of *Aspergillus niger* NRC1ami pectinase embedded in a polyvinyl alcohol (PVA) sponge, and the immobilization yield reached 66%. It was determined that the enzyme activity remained at the maximum when the sponge size was 20 mm × 20 mm × 2 mm, and the enzyme was immobilized at a concentration of 30.5 mg protein/g of support [14]. Rajdeo et al. [52] adsorbed pectinase on a polymer matrix through ion exchange. The results showed that the immobilized enzyme could be recycled more than 10 times, and the loss of its activity during apple juice clarification was less than 5%. Andrade et al. [54] studied a new tannase enzyme from *Penicillium rolfsii* CCMB 714 that was immobilized on calcium alginate. The immobilization yield reached 99.5%, showing a significant improvement in the physical and chemical stability of the enzyme.

#### 2.1.3. Covalent Binding

Covalent binding methods involve the use of support surface groups and active groups on the enzyme surface to form a strong covalent bond for the purpose of immobilization. The covalent bond is mainly formed by the strong interactions between amino (-NH2), carboxyl (-COOH), or sulfhydryl (-SH) groups on the surface of the enzyme and active groups such as aldehyde groups on the surface of the support. For example, the chlorine atom of cyanuric chloride can be replaced with different nucleophilic groups (-SH, -NH_2_, and -COOH) from the side surface of the protein to form stable bonds [55]. By using the immobilization method of strong covalent binding, the leakage of the enzyme can be significantly reduced. Kharazmi et al. [56] immobilized xylanase on magnetic nanoparticles grafted with trichlorotrigine-functionalized polyethylene glycol with a covalent binding method. The prepared immobilized xylanase had improved stability and increased catalytic activity. The capacity of the immobilized xylanase reached 260 mg of protein/g of support, showing its superior potential for clarifying pineapple juice [56]. One of the most commonly employed tools for achieving the covalent immobilization of an enzyme is the utilization of glutaraldehyde, which is usually achieved by employing aminated supports activated with glutaraldehyde [57]. Mohammadi et al. [58] used the active functional aldehyde group of glutaraldehyde to form covalent bonds with the amino group of montmorillonites and the primary amino group of the enzyme to achieve the immobilization of the enzyme. The immobilized enzyme showed strong stability and catalytic activity, and a remarkable effect on the clarification of pineapple juice was achieved.

#### 2.1.4. Cross-Linking

The cross-linking method is also a chemical immobilization method. The cross-linking method involves the use of bifunctional cross-linking reagents to connect enzyme molecules by forming covalent bonds to complete the immobilization of the enzyme. In the process of cross-linking, the role of the cross-linking agent is very important because it is related to the stability and activity of the enzyme. At present, glutaraldehyde [59] and dextran polyformaldehyde [60] are commonly used as crosslinking agents in immobilized enzymes for juice clarification. Glutaraldehyde is toxic; if it is leached from an immobilized enzyme, it will be unsafe for juice. At present, the probability of glutaraldehyde being used in juice clarification is decreasing. As a kind of polysaccharide cross-linking agent, dextran paraformaldehyde has attracted more and more attention and has become a research hotspot in recent years. Sojitra et al. [60] used dextran polyoxymethylene to immobilize pectinase and optimized the cross-linking parameters, such as the concentration of the cross-linking agent, the cross-linking time, and the ratio of the cross-linking agent to the enzyme. When the concentration of the cross-linking agent and cross-linking time reached 2.5% (*v*/*v*) and 15 h, respectively, the enzyme activity was the highest. When the two were further increased, the enzyme activity had a downward trend. When the ratio of the crosslinking agent to the enzyme was 4:1, the best immobilized pectinase activity was achieved [60,61]. The enzyme immobilization methods used for juice clarification are summarized in Table 1.

### 2.2. Immobilization Support

In the preparation of immobilized enzymes, in addition to choosing a suitable immobilization method, it is also very important to find the appropriate support for the immobilized enzyme (Figure 2) [81,82]. The immobilizing support plays an irreplaceable role in the immobilization of an enzyme and its application in fruit juice, and the choice of an immobilizing support is particularly important. Generally speaking, the immobilizing supports used for juice clarification need to meet the following four requirements: good thermal stability and physicochemical stability, low price and high safety for use in juice, good biocompatibility (which is conducive to the catalytic reaction of the), and high loading capacity (which does not lead to enzyme denaturation and inactivation). Magnetic nanoparticles [83], alginate [84], and chitosan [85] are commonly used as supports for immobilized enzymes in juice clarification. In recent years, researchers have also begun to study some new immobilization supports, including gelatin hydrogel [66], montmorillonite [78], and salinized glass beads [76].

#### 2.2.1. Magnetic Nanoparticles

Magnetic nanoparticles can be easily separated from a reaction medium under the condition of an external magnetic field, and their specific surface area is large, so they can load a large number of enzymes. Magnetic nanoparticles possess some advantages, such as their simple preparation, cost-effective synthesis, unique size, large specific surface area, high loading capacity for enzymes, low toxicity, chemically modifiable surface, biodegradability, biocompatibility [86], and ready dispersion in the aqueous phase [84]. Therefore, they are widely used in the immobilization of enzymes and have become potential supports. The main magnetic nanoparticles used are magnetic iron oxide particles. In most cases, there are some inherent limitations to working at the nanoscale, and nanoparticles are often prone to clumping together, leading to an increase in the size of the biocatalyst and thus a decrease in catalytic efficiency [32]. These limitations are often difficult to deal with at an industrial level. From an industrial perspective, the use of magnetic micro- and macrosubstrates as universal supports for immobilized enzymes can lead to more uniform biocatalysts, which also include better repeatability and lower manufacturing costs [32]. Another interesting advantage of magnetic micro- and macro-scaffolds compared to nano-magnetic scaffolds is zero or very low polymerization, which is easy to handle and recycle [87]. However, careful consideration should be conducted before the rational application of magnetic biocatalysts. Generally, the immobilized enzymes were used in heterogeneous reaction media and the nonmagnetic immobilized biocatalysts may not be easy to recover and recycle [32]. In these cases, magnetic supports and external magnetic fields can be applied [88]. A poor consistency of crosslinked enzyme polymers (CLEAs) occurs in aqueous systems. However, improved handling of magnetic CLEAs could be achieved using a magnet when they were prepared by co-precipitating enzymes and magnetic nanoparticles [89]. Similarly, enzyme nanoflowers provide good performance in enzyme immobilization, but they are difficult to recycle because of their small size. In contrast, magnetic nanoflowers are easy to be separate, and they provide a stable structure and improved catalytic efficiency [32,90]. In one-pot cascade reactions catalyzed by biocatalysts, the inactive enzyme can be easily removed if magnetic support was used for enzyme immobilization. For the enzymes produced by thermophilic microbes, the magnetic support can exhibit the local hyperthermia effect induced by magnetic fields, resulting in enhanced enzyme activity without destroying the substrate’s stability [32].

Mosafa et al. [69] used a co-precipitation method for the first time to covalently immobilize papain on magnetic nanoparticles coated with calcium silicate, and there was no phase change during the binding process of Fe_3_O_4_. Compared with the free enzyme, the enzymes immobilized on magnetic Fe_3_O_4_ nanoparticles (MNPs) exhibited improved catalytic activity, enhanced stability, and simple reusability [80]. Between the enzyme and the support, the covalent bond was stable enough to resist conformational changes and reduce the dissociation rate of the enzyme. The immobilized papain retained 75.1% of its activity after six catalytic cycles. Shahrestani et al. [72] successfully synthesized 1,3,5-triazine-functionalized silica-encapsulated magnetic nanoparticles and used them for the immobilization of xylanase. The immobilized xylanase showed excellent catalytic activity and maintained good stability when used for juice clarification. After 10 cycles of use, it retained about 55% of its activity. It was also found that some metal ions, such as Mn^2+^ and Mg^2+^, could effectively increase the activity of immobilized enzymes, and some organic solvents, such as acetone, toluene, and acetonitrile, could also increase this activity at a concentration of 10%. Sojitra et al. [79] covalently immobilized three enzymes (cellulase, pectinase, and amylase) on magnetic nanoparticles at the same time, which enhanced the tolerance of the immobilized enzymes to lower pH, and they showed high stability. After eight consecutive uses, the immobilized enzymes retained 75% of their activity. Dal et al. [74] prepared a multifunctional magnetic nano-starch catalyst by using magnetic iron oxide nanoparticles. The highest immobilization yields were 87% for pectinase, 69% for xylanase, and 58% for cellulase. At the same time, compared with the free enzymes, the immobilized enzymes retained much more of their activity in some organic solvents such as chloroform. Wang et al. [28] immobilized laccase on metal-chelated magnetic silica nanoparticles; the immobilization yield was 98.6%, and the activity of the immobilized laccase after storage for 10 weeks was 95.1% of the initial activity.

#### 2.2.2. Alginate

Alginate also known as alginate gel is a kind of polysaccharide extracted from seaweed plants; it is non-toxic, has mild gelling properties [50,84], and is a natural polymer material [91]. At present, sodium alginate and calcium alginate are commonly used in research [15,24,62,68]. Because of their wide ranges of sources, low prices, and good biocompatibility, they are currently widely used in the preparation of immobilized enzymes [20].

Bhushan et al. [15] found a simple immobilization method for encapsulating xylanase in alginate gel to enhance its resistance to heat inactivation. The storage stability of the immobilized enzyme was increased, and its activity remained at 80% of the original after one month of storage at 4 °C. It had better tolerance in the acidic range and is of great value for the processing of juice with a high pH [15]. Amin et al. [62] immobilized extracellular polygalacturonase on sodium alginate through covalent immobilization and adsorption methods. Both methods improved the catalytic activity and thermal stability of the enzyme. At the same time, they found that the covalent immobilization method was more effective than the adsorption method. After seven consecutive reaction cycles, the enzymes immobilized with the two methods retained 50.0% and 41.0% of their original activity, respectively. Oliveira et al. [53] immobilized pectinase on alginate microspheres and found that with a central composite rotatable design, the immobilization parameters, such as the concentrations of sodium alginate and calcium chloride, were optimized to achieve the best immobilization of the enzyme. The maximum immobilization yield was 56.71%. The immobilized pectinase had good operational stability, and its activity remained at 80% after three cycles of use [66]. Deng et al. [68] immobilized polygalacturonase on calcium alginate microspheres using the endogenous emulsification method. The immobilized enzyme showed good operational stability and retained 63% of its initial activity after three cycles. In addition, studies have shown that using calcium alginate as an immobilized support can allow good thermal stability to be maintained after four consecutive cycles [92].

#### 2.2.3. Chitosan

Chitosan is a linear polysaccharide comprised of connected β-(1,4)-glucos-amine; it is commercially acquired via the deacetylation of chitin from lobster and shrimp shells [24], and its molecular structure contains active amino groups and hydroxyl groups, which play an important role in immobilized enzymes and can be used as chemical modification sites for reactions [85]. Chitosan is widely used as an important support for enzyme immobilization due to its unique structure, cost effectiveness, and safe functional properties [93].

Irshad et al. [94] found that chitosan is a good immobilized support that can significantly improve the catalytic activity of enzymes, can increase their pH stability and thermal stability, and can be an ideal candidate material for the juice clarification industry. Benucci et al. [10] successfully immobilized two food-grade enzymes on chitosan microspheres and optimized the immobilization procedure. When the immobilization solution contained 1.0 mg of BSA eq/mL of protease and 1.8 mg of BSA eq/mL of pectinase, the immobilization effect was the best. Pomegranate juice was treated with the immobilized multi-enzyme system, which achieved a significant effect and retained the natural characteristics of the juice well. Dal Magro et al. [25] used chitosan as a support to immobilize an enzymatic cocktail named Novozym 33095, which included pectinase, polygalacturonase, pectinlyase, pectin methyl esterase, and cellulase. The stability of the immobilized enzyme was better than that of the free enzyme, and the catalytic activity was significantly improved after immobilization. Orange juice was clarified in a fluidized bed reactor, and 60% of the initial clarification ability was retained.

#### 2.2.4. Other Supports

In addition to the above three commonly used immobilized supports, other immobilized supports used in current research include pumice [48], salinized glass beads [91], montmorillonite [58], fiber [59], diatom soil [48], polyvinyl alcohol [14], agarose [95], monoaminoethylnethyl agarose ionophores [96], alumina microspheres [72,79], and some combinations, such as alginic acid/montmorillonite [78], alizarin sodium/graphene oxide composite beads [66], etc.

#### 2.2.5. Summary and Discussion

Although many studies have reported different immobilizing supports for enzyme immobilization in juice clarification, their large-scale application is still limited due to their high cost and insufficient operating half-life, as well as the low activity and stability of the enzymes immobilized on these supports. The application of supports such as alginate and chitosan can prolong the service life of enzymes and improve their stability. In addition, the immobilization process is simple, and the materials are cheap. However, these materials have poor mechanical properties and can be reused less frequently. Magnetic nanoparticles with stable physical and chemical properties can be separated and recovered from a reaction medium by using an external magnetic field, but magnetic nanoparticles are more expensive than sodium alginate, chitosan, and other supports. The fabrication of cheap magnetic supports with less aggregation can be an alternative method. In addition, researchers have also attempted to prepare composite supports, which can not only reduce the cost, but also can provide better stability than that of a single support.

### 2.3. Catalytic Conditions of Immobilized Enzymes in Juice Clarification

In juice clarification, the catalytic activity of the immobilized enzyme is affected by many factors, including the pH, temperature, enzyme dosage, and reaction equipment. The catalytic activity of the enzyme changes with these factors.

Each enzyme has an optimal pH value, so different pH values affect the recovery of enzyme activity. Mohammadi et al. [58] studied the catalytic activity of immobilized pectinase at different pH values (3–8) and found that with the increase in pH, the catalytic activity of the immobilized pectinase showed a trend of first increasing and then decreasing. At a pH of 5, the activity reached its maximum. At a pH of 3.0 and 4.0, the enzyme activity retained 60 and 79.2% of its initial activity, respectively, and this activity was significantly higher than that of the free enzyme, indicating that the stability of the enzyme under acidic conditions could be improved by immobilization. Most juice is acidic, so this enzyme is conducive to the clarification of acidic juice. Kharazmi et al. [97] studied the activity of immobilized enzymes at different pH values (3.5–8.5). The results showed that the activity of immobilized enzymes was the best at a pH of 4.5, and the activity of the immobilized enzymes was significantly higher than that of free enzymes at a pH less than 4.5, which would also be beneficial for the clarification of acidic juice.

Temperature is also an important factor affecting the catalytic activity of immobilized enzymes. Immobilization technology improves the thermal stability of enzymes, but higher temperatures can easily denature enzymes and greatly reduce their catalytic activity. Therefore, in juice clarification, it is very important to choose a suitable catalytic temperature. The activities of free xylanase and immobilized xylanase were measured at different temperatures (37–75 °C). However, the activity of the immobilized enzyme increased with the increase in temperature until a maximum of 65 °C and then decreased at 70 °C and 75 °C. The optimal temperatures for the free and immobilized enzymes were 60 °C and 65 °C, respectively, as the highest activities were achieved at these temperatures [70]. Mohammadi et al. [58] studied the catalytic activity of immobilized pectinase at different temperatures (30–60 °C), and they found that the catalytic activity of the immobilized enzyme was the best at 40 °C; then, the activity decreased as the temperature increased. However, the downward trend was slow, and good thermal stability was maintained. A greater kinetic energy of the molecule and a higher activity of the enzyme increase the possibility of the enzyme binding with the substrate. However, when the temperature exceeds the optimal limit, the support cannot protect the enzyme’s structure from deformation [58].

In the process of juice clarification with immobilized enzymes, the reaction equipment is also an important factor affecting the catalytic activity of the enzymes. The main reactors used for performing the reaction with the immobilized enzyme are stirred tank reactors, fluidized bed reactors, and packed bed reactors (Figure 3). Packed bed reactors are the most commonly used reactors for the continuous reaction of immobilized enzymes; they can be easily constructed and scaled up to improve the stability of biocatalysts [41]. Hosseini et al. [91] prepared immobilized pectinase on silicide glass microbeads with cross-linking agents and used the immobilized enzyme for pomegranate juice clarification in a packed bed reactor. It was found that the turbidity, viscosity, total soluble solids (TSSs), and pH of the juice could be significantly reduced in order to clarify it when the treatment was carried out at 50 °C for 42 min with a flow rate of 0.5 mL/min. It was shown that the maximum activity of the pectinase enzymes immobilized by the mentioned cross-linkers was obtained at 50 °C. Ozyilmaz et al. [2] used four different co-immobilization methods to co-immobilize amylase, pectinase, and cellulase on silica gel, and they applied these enzymes for the clarification of apple, grape, and pear juice in a packed bed reactor. The results showed that the reducing sugar concentration increased and the clarity was improved, indicating that the co-immobilization of amylase, pectinase, and cellulase exhibited good performance in juice clarification. The immobilization efficiencies of amylase, pectinase, and cellulase reached 67.9%, 53.6%, and 72.9%, respectively. Dal Magro et al. [25] studied the immobilization of a commercial enzyme mixture on glutaraldehyde activated chitosan beads and tested the resulting immobilized enzymes’ performance in continuous juice clarification in packed bed and fluidized bed reactors. The results showed that when the flow rate was 0.5 mL/min, the efficiency of apple juice clarification in a fluidized bed reactor was increased by 25% compared to that in a packed bed reactor because of the better mass transfer in the fluidized bed reactor. When its activity was measured at 90 °C and a pH of 4.8, the immobilized enzyme cocktail retained 80% of its initial activity, while the free enzyme retained only 35%. When the flow rate increased, the clarification abilities displayed a downward trend because the high flow rate meant a short retention time, resulting in less reaction time. It was also possible that more insoluble juice particles accumulated in the reactor at high flow rate and the substrate could not access the biocatalysts coated by these insoluble particles [25]. A low flow rate can allow enough pectin to diffuse into the internal area of an immobilized biocatalyst, reach the catalytic site of the enzyme, and provide good clarification [81]. Wang et al. [28] immobilized laccase on Cu^2+^-chelated magnetic silica nanoparticles to clarify fruit juice by using a magnetically stabilized fluidized bed with the aid of an alternating magnetic field. The use of the magnetically stabilized fluidized bed increased the mass transfer in the enzymatic reaction and improved the catalytic efficiency of the magnetically immobilized laccase. The application of an alternating magnetic field greatly improved the reaction rate and degradation rate of catechol with immobilized laccase in comparison with the rates under static and mechanical stirring conditions. No activity loss occurred in immobilized laccase after 20 h of continuous operation of juice treatment in MSFB under an alternating magnetic field. When microfiltration was also used after treatment with immobilized laccase, the color of apple juice decreased by 33.7%, and the light transmittance was enhanced by 20.2%. Furthermore, only 16.3% of the phenolic compounds and 15.1% of the antioxidant activity were reduced for apple juice after the clarification. As a result of these combined strategies, the apple juice possessed good freeze–thaw stability and thermal stability.

## 3. Application of Different Immobilized Enzymes in Juice Clarification

To improve the juice quality by green technologies to meet the increasing demand of natural fruit juices, the development of clarification technology using immobilized enzymes has attracted widespread attention [83]. Some components of the fruit itself, including pectin, starch, cellulose, and hemicellulose, are the main reasons for the turbidity and precipitation of fruit juice during juice processing and storage, leading to inferior quality of the fruit juice, greatly reducing its appearance and quality, and affecting its final shelf life and consumer awareness thereof [98]. In order to reduce or even avoid the impact of this phenomenon on the quality of juice and stabilize it, the application of enzymes in fruit juice clarification has attracted attention, especially for immobilized enzymes. Immobilized enzymes have shown excellent performance in fruit juice clarification (Table 2). Among these enzymes, pectinase is currently one of the most effective enzymes used in juice clarification. Secondly, laccase and xylanase have also shown good efficacy in juice clarification; they significantly improve the clarification of fruit juice, and they have increased in popularity.

### 3.1. Pectinase

At present, pectinase preparations have been immobilized using several supports, such as alginate [53], magnetic particles [56], and carbon (various agro-industrial residues and products) [47] to improve their catalytic performance in juice clarification and pectin hydrolysis. Pectin is a complex carbohydrate found in the cell walls of plants and the intercellular layer of plants [86]. Depending on pectin’s source, extraction methods, and processing, it is possible to release galacturonic acid, rhamnogalacturonans, and pectin oligosaccharides [105]. The content of pectin in fruits is low, especially in apples and citrus. Pectin is a polysaccharide with a high water storage capacity, and it can create a cohesive network structure in juice, which is also the main cause of juice turbidity. Therefore, it is particularly important to use immobilized pectinase to hydrolyze pectin when clarifying juice. Pectinase mainly hydrolyzes the α-1,4-glycosidic linkages in pectin [48,106] and depolymerizes and de-esterifies pectin; it degrades it to produce D-galacturonic acid and methyl galacturonate. It destroys the network structure of pectin and reduces its water-holding capacity. This enzyme can release free water into juice [65,107], which reduces its viscosity and turbidity and results in a better clarification rate.

Mohammadi et al. [77] used alginate montmorillonite microspheres as supports to covalently immobilize *Aspergillus* acanthus pectinase during the clarification of pineapple juice. The results showed that the initial activity of the immobilized enzyme remained at about 53% after six cycles, and the viscosity of the juice decreased by 40%. Oliveira et al. [53] immobilized the pectinase of *Aspergillus* acanthus in alginate beads and carried out a continuous clarification process in a packed bed reactor. The results showed that the protein (biocatalyst) concentration varied from 0.2 mg/mL to 0.7 mg/mL. The best immobilization yield was recorded at protein concentration of 0.5 mg/mL, and the clarification rate reached 97.22%. Oktay et al. [78] immobilized alkyne pectinase (≥3800 units/mL) on frozen polyvinylimide-based gel through a spontaneous aminogyne click reaction and then clarified apple juice. The results showed that the immobilization yield of alkyne pectinase was 90%, and the clarification rate reached 50%. Martin et al. [99] used the outer gel technology to immobilize enolated pectinase on agar–sodium alginate hydrogel beads to clarify grape juice. The results showed that the immobilized pectinase effectively decomposed pectin in grape pomace, and the light transmittance of the treated juice increased by 6.5 times compared with the untreated juice. The colors value decreased, the turbidity decreased rapidly, and 61% of the initial activity was maintained after six consecutive cycles. Bakshi et al. [101] used pectinase to clarify apple juice. The results showed that when lemon peel powder was used as an enzyme source, the enzyme activity was 2804.4 U/g, and the pectin was successfully hydrolyzed five times. The pectinase activity of the lemon peel powder decreased by 11.11% on the 60th day after storage at 4 °C. As a natural source of immobilized pectinase, lemon peel powder can effectively clarify apple juice. Chakraborty et al. [108] used the pectinase (polygalacturonase) enzyme from *Aspergillus niger* with a specific activity of 10 U/g for clarification. The clarity and yield were greatly enhanced by 44.7% and 43.4%.

The use of immobilized pectinase to clarify juice can prevent the hydrolysis of pectinase itself, and after processing the juice, the enzyme can be recovered with an external magnetic field, but there are also some problems. The use of immobilized pectinase to hydrolyze pectin leads to the deposition of some compounds, such as phenols, which will reduce their content in fruit juice, and this changes the distribution of phenolic compounds, which reduces the juice’s antioxidant capacity [66,99,109].

### 3.2. Laccase 

Laccase is widely used in fruit juice processing, as it can enhance the sensory quality of fruit juices, especially during juice clarification [110]. Laccases are multicopper proteins that catalyze the oxidation of various phenolic and non-phenolic compounds through the reduction of molecular oxygen via a radical-catalyzed reaction mechanism [111,112]; they are widely found in higher plants and most microorganisms, such as bacteria and fungi. The phenolic compounds in fruit juice are a class of healthy and beneficial compounds [113], but they also interact with proteins, making the juice turbid and affecting its appearance [114,115,116]. Laccase can oxidize the phenolic compounds in fruit juice to polymerize them [115,116] and form quinones, the precursors of brown pigment, and reduce oxygen to water [117,118]; these insoluble polymers can be separated through centrifugation. Some soluble polymers can be treated through filtration to improve the rate of fruit juice clarification.

Laccase has been successfully used to clarify apple juice, orange juice, pomegranate juice, and other products. For example, pretreated coconut husk fiber activated with glyoxyl or glutaraldehyde was fabricated and used as a support to immobilize 6.55 U of chemically modified laccase, and 59% activity recovery was achieved [59]. The results showed that the color value and turbidity of apple juice after immobilized laccase treatment decreased by 61% and 29% compared with those before the treatment. After 10 repeated uses, the laccase retained 100% of its initial activity and improved the sweetness of the apple juice. Recombinant POXA1b laccase was covalently immobilized on epoxy-activated polymethyl methacrylate beads to prepare immobilized laccase for the clarification of orange juice; the laccase activity reached 2000 ± 100 U/g of beads, with an immobilization efficiency of 98%, and the phenol in orange juice was reduced by up to 45% without affecting the healthy and effective flavanone content in the juice [73]. Laccase that was covalently immobilized on functional ferromagnetic nanoparticles was prepared with an initial concentration of 20 U/mL. The obtained immobilized enzyme was applied in the clarification of juices from banana pseudo-stems, sweet sorghum stems, and apple fruits; the turbidity, phenol content, and color value were reduced to 50–59%, 41–58%, and 49–59% of their initial levels, respectively [119]. Wang et al. [28] found that the use of immobilized laccase (from *T. versicolor*, ≥0.5 U/mg) combined with an alternating magnetic field to clarify apple juice with a magnetically stabilized fluidized bed reduced the color of the juice by 33.7% and the light transmittance by 20.2%, while the phenolic compounds were reduced by 16.3%, and the oxidation resistance was reduced by 15.1%. A significant clarification effect was achieved.

In juice clarification, immobilized laccase allows enzymes to be recovered and their reusability to be increased. In addition, it can increase the structural stability and catalytic stability of enzymes under acidic environmental conditions, such as those of juice, and it can prevent the product from covering the surface of an enzyme and reducing its catalytic activity. However, the immobilization of laccase also causes a loss of laccase activity.

### 3.3. Xylanase

In recent years, more and more attention has been paid to the application of immobilized xylanase in juice clarification. The main function of xylanase is the depolymerization of xylan. Xylan exists in plant cell walls and is the main component of plant hemicellulose. It is a hybrid polysaccharide in which the main chain consists of multiple xylopyranosyl groups connected by β-1,4-glycosidic bonds [120], and a variety of different substituents are connected on the side chain: α-L-arabinofuranosyl, acetyl and glucuronic acid residues, etc. [121]. Xylan contains a large number of hydroxyl groups, is highly hydrophilic, and easily forms network structures, making it one of the main reasons for turbidity in juice [122,123,124]. Xylanase is not an enzyme but a class of enzymes, which includes β-1,4-endoxylanase, β-xylosidase, α-L-arabinosidase, α-D-glucuronidase, acetylxylanase, and phenolesterase. The degradation of xylan is accomplished by multiple enzymes of this enzyme system. First, β-1,4-endoxylanase catalyzes the hydrolysis of β-1,4-glycosidic bonds in xylan molecules and hydrolyzes xylan into small oligosaccharides, xylobiose, and other xylo-oligosaccharides, as well as small amounts of xylose and arabinose. β-xylosidase can further hydrolyze the ends of xylooligosaccharides and then catalyze the release of xylose residues. Phenolicesterase can hydrolyze the ester bonds formed by arabinose side residues and phenolic acids in xylan. α-L-arabinosidase, α-D-glucuronidase, and acetylxylanase can act on the glycosidic bonds between xylose and the substituents of the side chain and can cooperate with β-1,4-endoxylanase to ultimately convert xylan into a mixture of xylose and xylo-oligosaccharides. After the degradation of xylan, the water-holding capacity of xylan decreases, and the turbidity and viscosity decrease [125]. The clarity of fruit juice obviously increases [95], and the reducing sugar content in the fruit juice increases [108,126].

Trichloro triazine-functionalized polyethylene-glycol-grafted magnetic nanoparticles were used for the preparation of covalently immobilized xylanase, and the turbidity of pineapple juice decreased by 42% after a 120 min treatment when the obtained catalyst was tested in juice clarification [56]. In addition, an upward shift in the optimal pH (6.5 to 7.5) and temperature (60 to 70 °C) of xylanase was observed after immobilization, and the performance of the immobilized enzyme was improved at high temperatures and pHs, as affirmed by the enhancement in vmax (2.69 to 6.01 U/mL). The immobilized xylanase retained about 50% of its initial activity after nine cycles of juice clarification [56]. Fe_3_O_4_@SiO_2_ nanoparticles were prepared as supports for the immobilization of xylanase (from *Thermomyces lanuginosus*, ≥2500 units/g); the immobilized xylanase retained about 55% of its initial activity even after 10 clarification reactions in concentrated fruit juice [71]. A crude enzyme mixture (protein content: 1.175 ± 0.032 mg/mL) containing cellulase (3.96 ± 0.32 U/mg protein), pectinase (3.31 ± 0.24 U/mg protein), and xylanase (2.90 ± 0.12 U/mg protein) was produced with *Aspergillus niger* and was immobilized on magnetic nanoparticles (MNPs) by cross-linking them with glutaraldehyde. In the clarification of papaya juice with the resulting immobilized enzymes, the light transmittance was increased by 64% and the reducing sugar content was increased by 26% [80]. Using immobilized xylanase to clarify fruit juice was significantly better than using free xylanase. The recyclability and reusability of immobilized xylanase are economical and effective for the juice clarification industry. Immobilized xylanase has improved heat resistance, long-term thermal stability, and operational usability, allowing it to meet certain industrial needs.

### 3.4. Other Enzymes

In addition to the commonly used immobilized enzymes described above, papain, tannase [64], invertase [65], amylase [80], manganese peroxidase [64], cellulase [80], ligninolytic enzyme [67], glucosidase [96], bromelain [10], etc., have been used for juice clarification. These immobilized enzymes have achieved good results. For example, Mosafa et al. [69] studied the performance of immobilized papain in pomegranate juice clarification. The protease was able to catalyze the hydrolysis of protein in the juice, thereby preventing the formation of complexes between positively charged proteins and negatively charged pectin [108]. Meanwhile, this also alleviated the formation of protein and phenolic compounds and effectively reduced the turbidity.

Jana et al. [64] studied the clarification of jamun and cashew apple juice with immobilized tannase. Immobilized tannase was applied in the clarification of jamun and cashew apple juice, where enzymatic treatment removed 60 and 51% of the tannin from the juice, respectively. Tannins are chelating, so they can chelate with carbohydrates, metals, and other substances. They can enhance the viscosity of fruit juice and increase its turbidity. The results showed that the clarification rate of the juice was increased by 5.2% and the color was reduced by 31.4% after the treatment with immobilized tannase. Bilal et al. [127] used immobilized manganese peroxidase to clarify apple juice and orange juice. Manganese peroxidase produced by *Ganoderma lucidum* has the ability to catalyze the oxidation of phenolic compounds in fruit juice and can reduce the turbid precipitation caused by interactions between these phenolic compounds and proteins. The immobilized manganese peroxidase reduced the color value of apple juice by 42.7% and its turbidity by 36.3%, and it reduced the color value of orange juice by 51.5% and its turbidity by 43.6%, which significantly improved the juices’ aroma and flavor. A heterologously expressed exo-polygalacturonase from *Sporothrix schenckii* was purified and immobilized on yolk-shell-structured magnetic mesoporous silica using the encapsulation and cross-linking methods, respectively. The two resulting magnetic immobilized enzymes (silica@Fe_3_O_4_/SsExo-PG and silica@Glu-Fe_3_O_4_/SsExo-PG) were further used for apple juice clarification [103]. The turbidity of the apple juice was reduced by 82% after a 1 h treatment for both biocatalysts and the immobilized enzyme retained 80% and 90% of its initial activity after 10 consecutive applications, respectively. 

### 3.5. Summary and Discussion

Enzymes are highly specific, and different enzymes can target different substrates. In juice clarification, different immobilized enzymes can be selected to remove different substances in juices that result in turbidity and instability. Although many immobilized enzymes have been prepared and applied for juice clarification, most research has focused on the immobilization of single enzymes. However, the catalytic abilities and efficiency of single enzymes are limited in terms of applications. In order to take advantage of different enzymes at the same time, the co-immobilization of multiple enzymes has also received extensive attention. Compared with the use of a single immobilized enzyme, the co-immobilization of multiple enzymes can be used to activate the synergistic effects of multiple enzymes, reduce the negative impacts of mul55tiple treatments with different substrates for single enzymes, and decrease the loss of nutrients in the process of juice clarification. However, the cost of doing so may be relatively high, and the technology is more complicated; furthermore, the safety problems caused by juice clarification should also be considered.

## 4. Conclusions and Prospects

The immobilizing support is a key component of an immobilized enzyme, and the support can affect the enzyme’s performance. To reduce the cost of immobilized enzymes, improving the reusability and immobilization performance of supports can be a viable method. It is also necessary to explore novel and cheap supports with a high adsorption capacity, high activity recovery, and high stability. The byproducts generated through agricultural production and processing are cheap and may be good candidates for supports. The size, structure, and surface functional groups of supports can influence their immobilization performance. Therefore, these indices should be considered during their selection and fabrication. The toxicity of supports for applications in juice clarification should also be evaluated. Enzymes are the real actors in the process of juice clarification. The substrate specificity and catalytic efficiency of an enzyme can be improved by using rational and irrational protein design or chemical modifications, resulting in enhancements in the catalytic abilities of immobilized enzymes. The development of novel enzymes with good resistance to extreme conditions is also recommended in order to improve the scope of application and catalytic efficiency of immobilized enzymes. Different methods of enzyme immobilization have advantages and disadvantages. Therefore, a compromise is made during the selection of an immobilization method; the link between enzymes and supports can be strengthened while ensuring a high activity recovery rate and good reusability. Because the immobilization in most methods is non-directional, the active site of an enzyme cannot be fully exposed to substrates due to the steric hindrance resulting from random connections between enzymes and supports. Thus, the site-directed immobilization of enzymes can be studied to improve the performance of immobilized enzymes in the future.

## Figures and Tables

**Figure 1 foods-12-04258-f001:**
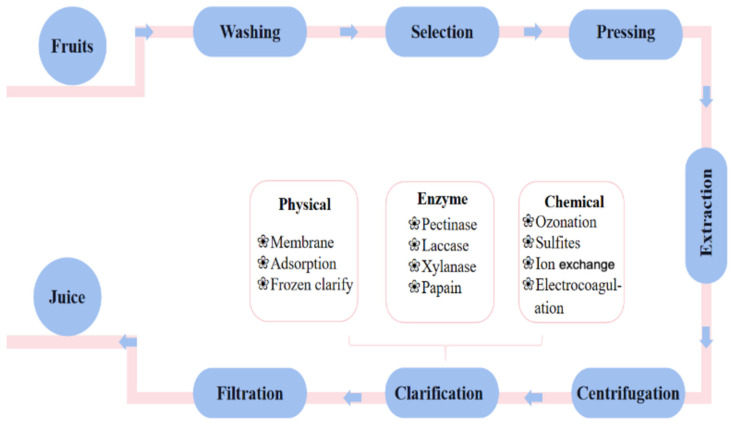
Flowchart of juice processing.

**Figure 2 foods-12-04258-f002:**
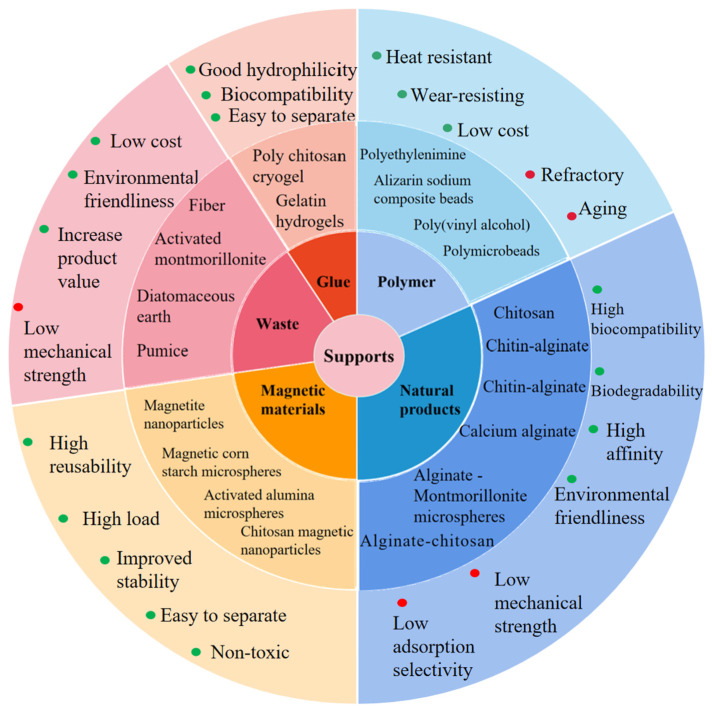
Supports for the immobilization of enzymes.

**Figure 3 foods-12-04258-f003:**
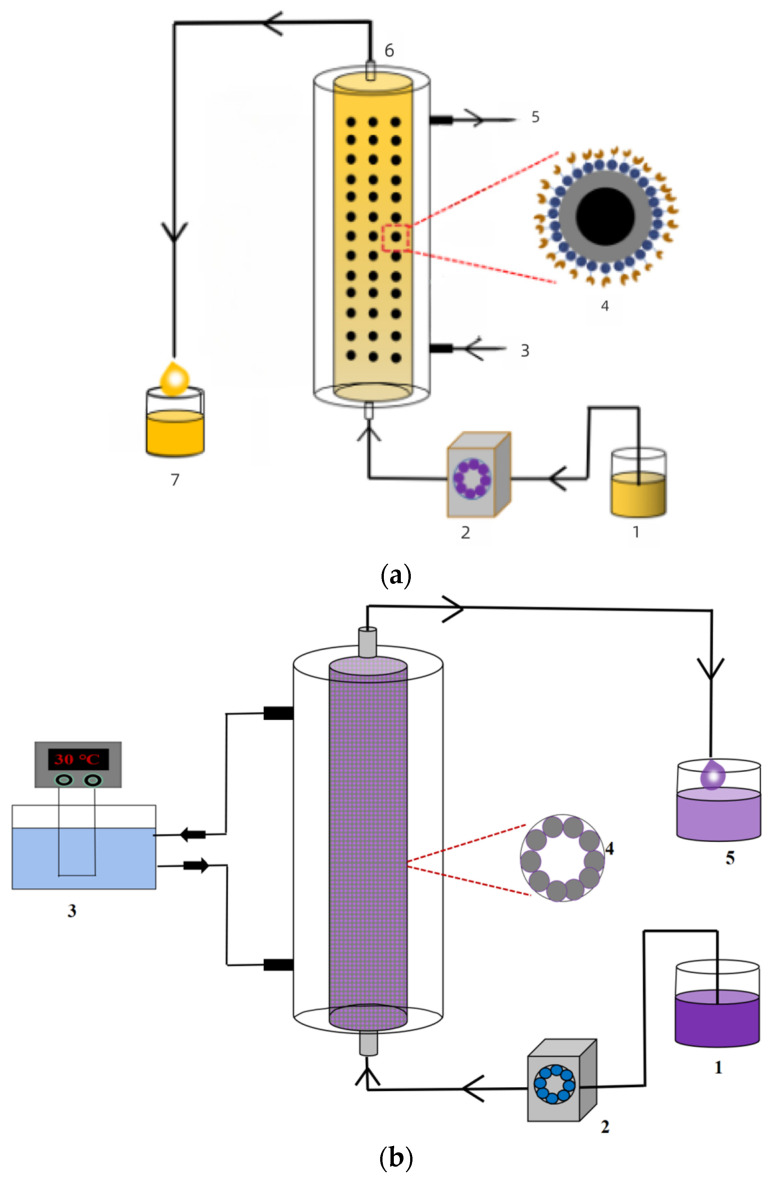
Bed reactors applied in juice clarification with immobilized enzymes. (**a**) Fluidized bed for clarifying juice: (1) fresh juice; (2) peristaltic pump; (3) water inlet; (4) immobilized enzyme; (5) water outlet; (6) fluidized bed; (7) clarified juice. (**b**) Packed bed reactor system for clarifying juice: (1) fresh juice; (2) peristaltic pump; (3) thermostatic bath; (4) immobilized enzyme; (5) treated juice.

**Table 1 foods-12-04258-t001:** Immobilization methods for different enzymes.

Immobilization Method	Support	Enzyme	Immobilization Capacity (mg/g Support)	Activity Recovery (%)	pH	Temperature (°C)	Ref.
Adsorption	Magnetic cornstarch microspheres (MMCSs)	Pectinase	50 *	60	4.5	50	[13]
	Celite	Polygalacturonase	20 *	/	5.5	45	[48]
	Polyethyleneimine-modified polymer	Pectinase	22 *	95	3.0	40	[52]
	Sodium alginate	Exo-polygalacturonase	/	/	5.5	55	[62]
	Zr-treated pumice	Pectinase	229	/	7.0	50	[49]
Entrapment	Polyvinyl alcohol sponge	Pectinase	30.5	91	6.0	50	[14]
	Calcium alginate beads	Pectinmethylesterase	/	/	8.5	60	[63]
	Alginate	Tannase	/	79	/	/	[64]
	Calcium alginate	Invertase	/	/	4.0	50	[65]
	Alginate beads	Xylanase	/	80	5.0	55	[15]
	Gelatin hydrogel	Manganese peroxidase	/	/	6.0	60	[66]
	Alginate chitosan	Ligninolytic enzyme	/	/	/	/	[67]
	Alginate beads	Pectinase	/	80	3.0	40	[53]
	Sodium alginate/Graphene oxide composite beads	Pectinase and glucoamylase	/	/	4.0	40	[66]
	Chitosan beads	Enzyme cocktail	300	/	4.0	70	[25]
	Calcium alginate	Tannase	/	50	4.3	44	[54]
	Calcium alginate microspheres	Pectinases	39	63	3.5	60	[68]
	Chitin + alginate	Tannase	/	/	5.0	40	[64]
	Charcoal + alginate	Tannase	/	/	/	/	[64]
Covalent binding	Magnetic nanoparticles (3-chloropropyl) tri-methoxysilane	Papain	/	~75.15	7.0	50	[69]
	PVP-stabilized ferrite-based silica-coated nanoparticles (Fe_3_O_4_–SiO_2_); functionalizing agent: (3Chloropropyl) trimethoxysilane	Papain	/	75%	9.0	80	[69]
	Magnetic nanoparticles Glutaraldehyde	Pectinase	19	/	4.0	50	[20]
	Aluminum oxide pellets	Extracellular xylanase	/	58	9.0	65	[70]
	Green coconut husk fibers Glutaraldehyde	Laccase	/	100	6.0	60	[59]
	Glyoxyl–agarose	Polygalacturonase	/	/	/	40	[71]
	Polyethylenimine–agarose	Polygalacturonase	/	/	/	/	[71]
	Monoaminoethyl–N-aminoethyl–agarose	Polygalacturonase	/	/	/	/	[71]
	Magnetic nanoparticles	Xylanases	280	55	6.5	65	[72]
	Poly(methacrylate) beads	Laccase	/	67	6.5	75	[73]
	Sodium alginate	Exo-polygalacturonase	/	/	5.5	55	[62]
	Iron oxide nanoparticles	Pectinase	/	/	/	/	[74]
	Iron oxide nanoparticles	Xylanase	/	/	/	/	[74]
	Iron oxide nanoparticles	Cellulase	/	/	/	/	[74]
	Magnetite Glutaraldehyde	Pectinase and cellulase	/	10	3.0	60	[75]
	Glass beads Glutaraldehyde	Pectinases	/	~59	5.5	50	[76]
	Magnetic nanoparticles	Pectinase	173	/	4.5	50	[56]
	Magnetic nanoparticles	Xylanase	/	/	7.5	70	[56]
	Cryogels	Papain	15.2 ± 2.54	/	8.0	65	[29]
	Magnetic chelator nanoparticles	Laccase	/	/	5.0	50	[29]
	Aluminum oxide pellets Glutaraldehyde	Multi-enzymatic system	/	/	5.0	60	[77]
	Alginate–montmorillonite beads	Pectinase	/	/	5.0	40	[78]
	Polyethyleneimine cryogel	Pectinase	/	/	6.5	55	[79]
	Montmorillonite Glutaraldehyde	Pectinase	/	60	5.0	40	[58]
Cross-linking	Magnetic nanoparticles Glutaraldehyde	α-Amylase, pectinase and cellulase	/	75	6.0	50	[80]
	Chitosan magnetic nanoparticles Dextran polyaldehyde	Pectinase	/	85	/	/	[60]
	Fe_3_O_4_ magnetic nanoparticles Glutaraldehyde	Cellulase, pectinase and xylanase	/	/	~5.0	60	[80]
	Oxides: ferrites; functionalizing agent: 3-amino propyltriethoxysilane (APTES) Glutaraldehyde	α-Amylase, pectinase, and cellulase	/	77	5.5	45	[79]
	Ferrite-based nano-particles; functionalizing agent: 3-amino propyltriethoxysilane (APTES) Glutaraldehyde	Cellulase, pectinase, and xylanase	/	85	4	50	[81]
	Ferrite-based nano-particles; functionalizing agent: 3-Amino propyl-trimethoxy silane (APTMS) Glutaraldehyde	Pectinase and cellulase	/	10	4.8	60	[74]

/: Not determined; *: Calculation according to the data from the literature.

**Table 2 foods-12-04258-t002:** Conditions and effects of juice clarification.

Immobilized Enzyme	Immobilization Method	pH	Temperature (°C)	Time (h)	Reaction Equipment	Source of Fruit Juice	Clarification Rate (Increased)	Turbidity (Reduced)	Viscosity (Reduced)	Color (Reduced)	Transmittance (Increased)	Reducing Sugar (Increased)	Ref.
Enzymatic cocktail	Entrapment	4.8	40	71	Fluidized-bed/packed-bed reactor	Orange	83%	/	/	/	/	/	[25]
Enzymatic cocktail	Entrapment	4.8	40	54	Packed-bed reactor	Orange	87%	/	/	/	/	/	[25]
Tannase	Entrapment	/	30	2.0	Shake flask	Apple	/	70%	44.7%	/	/	/	[54]
Xylanase	Covalent	4.5	50	2.0	/	Pineapple	/	42%	/	/	/	/	[56]
Laccase	Covalent	/	35	0.5	Fluidized bed	Apple	/	/	/	33.7%	20.2%	/	[29]
Pectinase	Covalent	3.4	40	3.0	/	Pineapple	/	/	40%	/	84.5% *	20.7%	[78]
Bromelain, pectinex	Covalent	/	30	8.0	Fluidized bed	Pomegranate	/	35%	/	/	/	/	[10]
Pectinase	Covalent	/	20	2.5	/	Grape	/	/	/	62.2% *	554% *	/	[99]
Papain	Covalent	/	50	1.0	Shake flask	Pomegranate	/	51.7% *	/	/	/	/	[69]
Pectinase	Adsorption	4.5	50	1.5	/	Apple	/	/	/	/	/	/	[13]
Pectinase	Entrapment	3.5	50	24	/	Orange	/	/	75%	/	/	/	[14]
Pectinmethylesterase	Entrapment	/	4	1.0	/	Orange	/	/	56%	/	57.1%	/	[63]
Pectinase	Covalent	6.5	55	/	Shake flask	Apple	50%	/	/	/	/	/	[79]
Pectinase	Covalent	/	40	3.0	/	Pineapple	/	/	/	/	41.97% *	/	[58]
Pectinase	Covalent	4.5	50	2.0	/	Pineapple	/	59%	/	/	/	/	[97]
Pectinase	Cross-linking	/	50	0.7	Packed bed reactor	Pomegranate	18.46% *	75.2% *	/	/	/	/	[91]
A tri-enzyme mixture of cellulase, pectinase, and xylanase	Cross-linking	5.0	55	1.5	/	Papaya	/	/	/	/	9.36% *	198% *	[80]
Pectinase	Entrapment	3.0	40	0.83	Packed bed reactor	Apple	97.22%	/	20.8%	/	/	/	[53]
Exo-polygalacturonase	Covalent	/	50	1.0	/	Apple	/	72.7%	66%	/	/	/	[62]
Exo-polygalacturonase	Covalent	/	50	1.0	/	Grape	/	72.2%	85.5%	/	/	/	[62]
Exo-polygalacturonase	Covalent	/	50	1.0	/	Peach	/	86.4% *	84.3% *	/	/	/	[62]
Pectinase	Cross-linking	/	50	2.5	/	Apple	/	74%	/	/	/	/	[60]
Ligninolytic	Entrapment	/	/	1.0	Packed bed reactor system	Apple	/	84.02%	77.04%	/	/	/	[67]
Ligninolytic	Entrapment	/	/	1.0	Packed bed reactor system	Grape	/	57.84%	83.01%	/	/	/	[67]
Ligninolytic	Entrapment	/	/	1.0	Packed bed reactor system	Orange	/	86.14%	75.86%	/	/	/	[67]
Ligninolytic	Covalent	/	/	1.0	Packed bed reactor system	Pomegranate	/	82.13%	86.95%	/	/	/	[67]
A tri-enzyme mixture of cellulase, pectinase and xylanase	Covalent	/	45	0.17	/	Pineapple	/	/	/	/	30%	12%	[74]
A tri-enzyme mixture of cellulase, pectinase and xylanase	Covalent	/	45	0.17	/	Orange	/	/	/	/	29%	26%	[74]
Polygalacturonase	Cross-linking	/	/	1.0	/	Apples	/	75.5%	81% *	64%	/	/	[94]
Polygalacturonase	Cross-linking	/	/	1.0	/	Peach	/	68.4% *	80% *	61.8% *	/	/	[94]
Pectin methylesterase	Cross-linking	/	/	1.0	/	Mango		70% *	65% *	60% *	/	/	[94]
pectinlyase	/	/	/	1.0	/	Apricot	/	80% *	80% *	65% *	/	/	[91]
Pectinases	Covalent	/	40	2	Batch reactor	Apple	/	80%	/	/	/	/	[100]
Pectinases	Covalent	/	40	2	Batch reactor	Pomegranate	/	4%	/	/	/	/	[100]
Pectinases	Covalent	5.02	30	0.5	Lemon peel	Apple	/	43%	/	/	/	/	[101]
Enzymatic cocktail	Cross-linking	4.8	90	1.5	Packed bed and fluidized bed reactors	Orange	60%	/	/	/	/	/	[25]
A tri-enzyme mixture of cellulase, pectinase, and xylanase	Covalent	/	50	3	/	Apple, grape, and pear	73%, 67%, 57%	27%, 33%, 43%	/	/	/	/	[2]
Exo-polygalacturonase	Cross-linking	6.5	70	/	/	Grape and pineapple	93%	/	55%	/	/	/	[102]
Exo-polygalacturonase	Cross-linking	4	60	/	/	Apple	/	82.0%	/	/	/	90%	[103]
Pectinases	Cross-linking		50	0.5	Packed bed reactor	Pomegranate	96.13%	82.6%	/	/	/	/	[91]
Pectinases	Cross-linking	/	50	2.5	/	Apple	44.7%	/	/	/	/	/	[104]
Pectinases	Cross-linking	/	50	1.5	/	Orange	100%	65%	/	/	/	/	[104]
α-Amylase, pectinase, and cellulase	Cross-linking	3	50	2.5	/	Apple, pineapple, and grapes	41%, 46% and 53%	/	/	/	/	/	[79]

/: Not determined; *: Calculation according to the data from the literature.

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
