# Peer review of "Application of Immobilized Enzymes in Juice Clarification"

_foods, 2023, doi:10.3390/foods12234258_

Round 1

Reviewer 1 Report

Comments and Suggestions for Authors

I wonder if this is a mistake a really they mean this (in this case, explain) “. At present, the bienzymatic”, In that sentence, the enzymes must be in plural.

Free enzymes has another problem, they will be incorporated to the juice, and this can generate some problems (e.g., a percentage, even small, of consumers may be allergic to this enzyme, the reaction will not stop and the control is no easy).

Immobilization is not always via attachment to a support. Please, revise this (copolymers, CLEAs, CLECs, nanoflowers, crystals coated of proteins, do not involve a support).

Immobilization is the “star” of the review, and introduction is extremely poor. Objectives are no longer just the reuse and a better control of the reaction. A proper immobilization (multipoint or multisubunit) can greatly improve the enzyme stability, even being a difficult target. Moreover, immobilization may be coupled to purification of the enzyme. This is achieved only if the protocol is adequately designed. Moreover, just by random, immobilization can alter enzyme activity, selectivity, specificity, inhibitions, etc. There are many reviews on these matters (that are no cited), they should use some paragraphs to properly explain the objective of modern immobilization. A recent review from Prof Woodley´s group discusses if the enzyme immobilization is or is not (and this is the conclusion) mature discipline.

In this application, the substrates are very large. This poses some additional difficulties (proper enzyme orientation is critical, high substrate diffusion limitations). Moreover, the substrate and very likely the final product are a suspension. This is one of the applications of magnetic biocatalysts described in the r4view from Fernandez-Lucas, magnetic nanoparticles may solve many problems, but they have other problems, magnetic porous supports may be a good solution. Pros and cons of both alternatives should be discussed.

A short discussion on advantages and drawbacks of different immobilization strategies for this application must be mentioned. Physical adsorption means the possibility of enzyme release and adsorption of some components of the juice, but permit the support resuse after enzyme inactivation. Covalent immobilization permits stabilization via multipoint povalent attachment and prevent enzyme release, but after inactivation of the enzyme the support must be discarded. Coimmobilizatino of several enzymes have advantages and drawbacks.

Although the specific applications are more or less adequate, the failures (lack of information, mistakes)  in this preliminary introduction greatly affect the interest and relevance of the current review.  A review is expected to give a minimum of information difficult to gather reading specific papers, and be written by experts in the field able to make a real critical analysis of current perspective and future development. The summary of the papers on the field by itself is not a real review.

Perhaps in some points they can increase discussion on the summarized papers 

Comments on the Quality of English Language

It has some small matters, but in general, it  is readable.

Reviewer 2 Report

Comments and Suggestions for Authors

The ms reviews an important subject and is scientically sound  and comprehensive but the English language is in need of substantial improvement. The length could also be significantly reduced by removing a lot of unnecessary repetition. This would make the ms acceptable for publication.

Comments on the Quality of English Language

The quality of the English needs to be sustantially improved by a native English Speaker. In particular, it could be substantially shortened by removing a lot of the repetition in wording which is not necessary in English. The Tables could also be simplified to make them more readable. It would then be acceptable for publication.

Reviewer 3 Report

Comments and Suggestions for Authors

The paper describes methods of enzyme immobilization used for clarification of juices. In general, the topic is relevant, but the work is not well systematized and the information is not properly taken from the literature. Also, the English language should be reviewed in detail and the fluidity of the work should be improved. I do not recommend this review for publication.

Examples of objections:

In general: the parameters used for immobilization success should be standardized and defined. The term immobilization rate should be avoided as it is very rarely used in the optimization of the immobilization process and the author has often replaced this term with immobilization yield, which is not the same. In general, it is better to use capacity for the methods that use carrier for immobilization. The retained activity is also an important parameter that should be included. I know that sometimes there are misunderstandings about the terminology, but in all papers it is usually defined and thus it is possible to extract the necessary data.

The chapter 2.1. Immobilization method and chapter 3. Application of different immobilized enzymes in juice clarification should be systematically unified. Indeed, chapter 3 does not specify which immobilization method was used for the mentioned enzymes and process. This means that the reader still has to read all the cited articles, and that is not the purpose of the review.

1.       Title: Enzyme immobilization in juice clarification should be reformulated to e.g. Application of immobilized enzyme in juice clarification.

2.       Line 78 – immobilization is not a brand new technology

3.       Lines 99 – 103; 174-175; 183-185; 244; 403-405 – the sentence is not clear

4.       Figure 1 - The figure is not cited in the text.

5.       Lines 126-127 – encage method is not the term commonly used in enzyme immobilization, nor is the co-price combination method.

6.       Line 212 – The sentence is not necessary.

7.       Figure 2 - The figure is not cited in the text. The figure is generally not clear. Which immobilization methods are concerned?

8.       Table 1 – The retained activity must be added as a column

9.       Line 365 – The highest activity

10.   Line 372 – Why does the temperature increase when the activity decreases?

11.   Line 383-385 – Mohamadi et al do not mention immobilization rate in their paper, but immobilization yield.

12.   Lines 386-390 – The paragraph is unclear

13.   Lines 377 – 392 – do not belong in this chapter. This is actually about optimizing the immobilization method.

14.   Lines 392 - 423 – This paragraph can be a very promising chapter as it includes the application of immobilized enzymes in different reactor configurations. But very few examples are given in the chapter and there are also no details about the enzyme immobilization method used and the stability of the enzymes.

15.   Figure 5 - The purpose of this figure is not clear. It is also not cited in the text.

16.   Figure 6 – The figure is not cited in the text

17.   Table 2- Table is not cited in the text. It is also not clear whether these are immobilized enzymes and what method of immobilization is used.

Reviewer 4 Report

Comments and Suggestions for Authors

The authors gave the summary of enzyme immobilization methods with emphasis on their application in juice clarification. They described reaction conditions, selectivity and effectiveness of immobilized enzymes previously applied in juice clarification. Although this review is giving some valuable insights into the topic, it is full of both content and linguistic errors, therefore it should be substantially revised to improve its quality and accuracy.

Page 2. Raw 62 – did authors mean bienzymatic or bio-enzymatic? Also, typing from that sentence to the end of paragraph needs to be edited (spaces, comas, capital letters…). Last sentence of this paragraph needs to be rephrased to be more related to the previous one.

Page 2, line 70 – sentence needs to be rephrased, it doesn’t make sense. The next sentence (line 72) should be deleted since more detailed explanations of all benefits of immobilized enzymes application are given later on within the paragraph.

Page 2, line 78 – I recommend avoiding terms such as brand-new for the technology that dates back to the fifties.

Page 2, lines 98-100 – It is more suitable to say enzyme immobilization method and strategy than to repeat word immobilized and immobilization so many times. Also, enzymes are immobilized, not carriers, you should say enzyme immobilization carrier or immobilization support. Again, typing errors are present all through the rest of the paragraph, so entire manuscript should be carefully edited.

Page 3, line 110-114 – enzyme should be plural at all three mentions

Figure 1. – the authors did not refer to this figure in the text

Page 3, line 117 – it should be written: …defined as a…

Page 3, lines 121-126 – Why all of a sudden terms fixed and fixation appeared in this part of the manuscript? Be consistent and use terms immobilized and immobilization.

Page 3, lines 126 and 127 – Can authors provide some widely used synonyms for encage method and co-price combination method?

Page 3, line 132 - …through weak interaction combination… Define them.

Page 4, lines 142-149 – the authors should make distinction between ionic adsorption and ion exchange.

Page 4, line 157 – capital letter again appeared in the middle of the sentence after comma. Check entire manuscript for this kind of errors.

Page 4, line 164 - 30.5 mg protein/g carrier, was it initial (offered) protein concentration or immobilized one? Please define.

Page 4, lines 166-169 – Written text seems not to be related to the given reference.

Page 4, line 174 – sentence is unclear.

Page 4, lines 183-185 – references are missing.

Page 5, line 188 – mg of bound proteins per g of support is better to be named bound proteins or amount of immobilized proteins, immobilization rate could be misunderstood as immobilization process velocity.

Fig. 2 – the authors did not refer to this figure in the text. It seems that classification from that figure is not similar to one described in related subsection.

Table 1 – are all examples from the table applicable for juice clarification? Immobilization amount is not adequate term, it could be named in the same way as in the text (page 5, line 188). Again, typing errors should be corrected. The authors should add one more column with information about immobilized preparation activities and also in the column in which type of enzyme is defined more precise information should be given about exact preparations that were used.

Page 10, line 227 and 232– immobilization support instead of immobilized carrier. Last two sentences from that paragraph need references.

Page 10, line 244 - Magnetic nanoparticles are mainly used for magnetic iron oxide particles. What did authors mean by this statement?

Page 10, lines 249 and 250 - silicon-calcium-coated magnetite nanoparticles coated with calcium silicate. Isn’t it enough just to say magnetite nanoparticles coated with calcium silicate?

Page 10, lines 268 and 272 – the authors should use the same term as in table – immobilization yield instead of immobilization rate.

Page 11, line 291- I recommend using term rotational or rotatable instead of rotation

Page 11, line 306 – carrier in immobilized enzymes? Is it carrier for enzyme immobilization?

Page 11, line 308 – immobilization support instead of immobilized carrier.

Page 11, line 316 – which enzyme cocktail?

Page 11, lines 321 and 322 - immobilization support instead of immobilized carrier.

Page 12, line 365 and 366 - …The highest temperature was 65℃, and then decreased to 82% and 74% at 70℃ and 75℃, respectively… Highest enzyme activity was achieved at 65 ℃?

 Page 12, line 372 and 373 - …If the use time is too long, the activity will drop rapidly, and the temperature will increase… I don’t understand this, why would temperature increase due to prolonged use? Did you mean that at increased temperatures and prolonged usage fast activity decrease is expected?

Page 12, line 372 - …At this time, the carrier-binding enzyme is saturated… Unclear, rephrase.

Page 13, line 392 -  …In the clarification of immobilized enzymes… Did authors mean in the clarification of juices by immobilized enzymes?

Page 13, line 405 - …When the packed bed reactor was 0.5 mL /min… Did authors mean that flow rate was 0.5 mL /min?

Page 13, line 415 - Fig. 3 and Fig. 4 are two commonly used devices. I suppose that authors meant that these fig-s are representing schemes of two commonly used devices…

Page 14, line 433 – what authors meant by ”national enzymes”?

Figure 5 – the authors, again, did not refer to this figure in the text. Since it is not related to manuscript content I suggest removing it from the manuscript.

Page 14, line 446 – what kind of carbon? Punctuation marks in that and following sentence need revision.

Page 15, lines 460-477 – due to versatility of pectinases, the authors should include information about preparations used and present pectinolytic activities (where available).

Page 155, line 492 – what kind of fibers?

Page 15, line 484 - …and it has a good catalytic function and can catalyze various oxidation reactions… This part of the sentence is unnecessary, it should be deleted.

Page 15, line 497- it seems that the beginning of the sentence is missing.

Page 16, line 502 - What does it mean to ”reduce the clarification process”, please rephrase.

Pge 16, line 524-525 - The authors stated that: Xylanases are an important class of carbohydrate enzymes,  catalyzing the hydrolysis of β-1,4 glycosidic bonds between xylose units in xylan, however they are catalysing hydrolysis of other bonds, therefore this sentence should be removed and the next sentence is more informative and correct.

Pge 16, lines 540-555 – xylanolytic activities should be defined in more details. Line 548 – which three enzymes?

Figure 6 – the authors did not refer to this figure and it contains several mistakes, therefore it should be removed from the manuscript.

Page 17, line 572 - …enhance the viscosity of fruit juice, and improve the clarity… Please explain, increased viscosity would usually lead to increased turbidity, not clarity…

Table 2 – the authors did not refer to this table in the text.

Conclusions and prospect could more concise and focused on few most important conclusions and future perspectives.

Comments on the Quality of English Language

Extensive editing of English language is required.

Round 2

Reviewer 1 Report

Comments and Suggestions for Authors

The authors have performed a serious revision of the paper, however, some points require further attention.

Minor points

Revise references, some authors names seem to be wrong.

Comment 1. Bio enzymatic is redundant, enzymatic is enough.

Major points

Comment 4. The introduction to immobilization is weak yet. Make a deeper presentation on how immobilization can improve enzyme features if properly performed, stabilization of enzymes via immobilization may be achieved if the protocol is properly performed, purification of the enzyme the same, while changes in selectivity and specificity are more related to random changes in enzyme conformation. Use some of the reviews on these subjects as references. In a introduction on enzyme immobilization, this should cover in a deep way what is expected of immobilization.

Comment 5. Answer is not enough, as orientation is not considered, only recovery of immobilized enzymes on a suspension. Again, there are reviews on these matters.

Reviewer 3 Report

Comments and Suggestions for Authors

The comments are added directly to the pdf file
